# Towards a Unified Management Interface for 5G Sensor Networks: Interoperability between Yet Another Next Generation and Open Platform Communication Unified Architecture

**DOI:** 10.3390/s24196231

**Published:** 2024-09-26

**Authors:** Devaraj Sambandan, Devi Thirupathi

**Affiliations:** 1Department of Computer Applications, Bharathiar University, Coimbatore 641046, India; 2PSG Institute of Management, PSG College of Technology, Coimbatore 641004, India; deanresearch@psgim.ac.in

**Keywords:** Industry 4.0, OPC UA, YANG, 5G sensor network, private 5G

## Abstract

Fifth-generation (5G) sensor networks are critical enablers of Industry 4.0, facilitating real-time monitoring and control of industrial processes. However, significant challenges to their deployment in industrial settings remain, such as a lack of support for interoperability and manageability with existing industrial applications and the specialized technical expertise required for the management of private 5G sensor networks. This research proposes a solution to achieve interoperability between private 5G sensor networks and industrial applications by mapping Yet Another Next Generation (YANG) models to Open Platform Communication Unified Architecture (OPC UA) models. An OPC UA pyang plugin, developed to convert YANG models into OPC UA design model files, has been made available on GitHub for open access. The key finding of this research is that the proposed solution enables seamless interoperability without requiring modifications to the private 5G sensor network components, thus enhancing the efficiency and reliability of industrial automation systems. By leveraging existing industrial applications, the management and monitoring of private 5G networks are streamlined. Unlike prior studies that explored OPC UA’s integration with other protocols, this work is the first to focus on the YANG–OPC UA integration, filling a critical gap in Industry 4.0 enablement research.

## 1. Introduction

The advancement of wireless communication technologies has had a profound impact on the development and implementation of sensor networks. The emergence of private 5G networks is revolutionizing Industry 4.0 by introducing new wireless communication systems, allowing sensor networks to achieve unprecedented levels of connectivity, reliability, and efficiency by facilitating the interconnection of objects, machines, and individuals [1,2]. The private 5G market is experiencing rapid growth, driven primarily by Industry 4.0, which requires real-time data and reliable communication. According to a verified market research report, the global market size of private 5G deployments was valued at USD 3.92 billion in 2023, with a projected compound annual growth rate (CAGR) of 38.80% from 2024 to 2031 [3]. Similarly, the Internet of Things (IoT) sensor market, valued at USD 12.34 billion in 2023, is expected to grow at a CAGR of 31.5% from 2024 to 2032, as reported by Polaris market research [4]. The key characteristics of private 5G sensor networks include ultra-low-latency communication (URLLC), enhanced mobile broadband (eMBB), massive machine-type connectivity (mMTC), security and privacy, and network slicing [5,6]. Private 5G sensors provide ultra-low latency as low as 1 MS, which is crucial for time-sensitive sensor applications in industrial automation, where rapid data processing and response are required. Private 5G sensor networks provide significantly higher data rates than previous generations, enabling real-time transmission of large amounts of sensor data. Also, private 5G sensor networks can support up to one million sensors per square kilometer, making it ideal for dense sensor deployments in large industrial environments [7]. On-premises deployments of private 5G sensor networks provide control over security and data privacy and mitigate the risks associated with public networks [8]. The private 5G sensor networks allow for the creation of virtual networks tailored to specific applications or use cases, thus ensuring that sensor clusters operate efficiently under different requirements [9].

However, there are significant challenges to realizing private 5G sensor networks due to their deployment complexity and the multiple different hardware/software components from different vendors involved [10,11]. A verified market research report identifies interoperability issues as a critical obstacle when private 5G networks integrate with existing technologies and devices requiring seamless communication across various systems and standards. Achieving these demands requires meticulous planning and robust solutions, which can complicate deployment and lead to higher costs [3]. The market and market reports highlight the seamless integration of private 5G networks with legacy systems as another significant challenge. This integration requires thorough consideration of compatibility issues to ensure uninterrupted data exchange between legacy systems and private 5G networks [12]. The market and market IoT sensor research report also stresses the need to establish common protocols and communication standards to enable the interoperability of IoT sensor devices [13]. Polaris market research points out that security and privacy concerns are significant challenges for sensor networks [4]. A CIO survey by NTT reveals that the most common barriers to deploying private 5G networks are the integration of the technology with legacy systems and networks, the complexity of deploying and managing these networks, and a lack of technical skills and expertise among employees to manage them [14,15].

Given these interoperability and network management challenges, it is essential to develop solutions that facilitate the seamless integration of private 5G networks with legacy systems and ensure their manageability through existing industrial systems. Figure 1 illustrates a private 5G sensor network within an industrial automation environment, addressing interoperability and network management challenges. The systems’ components are outlined by boundaries and labeled A, B, C, and D. Within boundary A, industrial sensors and applications serve as network endpoints by collecting data and measuring environmental or operational parameters, which are then transmitted to a central platform for analysis. Boundary B represents the private 5G sensor network, which includes 5G radios (RU) mounted on the factory floor’s ceiling. These radios function as sensor base stations, converting radio frequency signals into internet packet (IP) signals, which are then relayed to the distributed unit (DU) via access/aggregator switches and gateways. The DU forwards the data to the centralized unit (CU), both of which are components of the 5G radio access network (RAN) deployed on-site in the factory. These units handle radio signal processing and facilitate high-speed, low-latency wireless communication between the sensors and the 5G core (5GC). The 5GC manages connectivity, mobility, and session control while also supporting service customization through features like network slicing. Boundary C encompasses edge computing infrastructure, which hosts sensor sink nodes and other Industry 4.0 applications. This setup enables data processing and analysis closer to the sensors, thus reducing latency and offloading computation from centralized cloud servers. Finally, boundary D houses unified industrial automation applications responsible for managing and monitoring Industry 4.0 devices and systems.

In existing industrial automation environments, the devices and applications in boundary A and C rely on the OPC UA protocol, a widely adopted standard for industrial system information exchange, to integrate with other applications, such as SCADA, MES, SAP, ERP, and cloud-based manufacturing systems [16]. Meanwhile, the components, like aggregator/access switches, time-sensitive networking (TSN) devices, virtual gateways/routers, 5G radios, CUs, and Dus, expose YANG data models for information exchange, as standardized by the Internet Engineering Task Force (IETF), the 3rd-Generation Partnership Project (3GPP), and Open RAN Working Groups 4 and 10 [17,18]. Due to the differing semantics of YANG and OPC UA, achieving interoperability between private 5G sensor networks and existing industrial-application management systems that use OPC UA is challenging. This article proposes an interoperability layer, identified in Figure 1 as the “Mapper” component, to address these challenges.

In the last two decades, several solutions dealing with the interoperability between the OPC UA information model and other protocols/models have appeared in the literature due to the critical role played by OPC UA in Industry 4.0 automation. However, a significant gap in the literature is the lack of existing work that addresses the interoperability of YANG with OPC UA. This research aims to fill the gap by proposing a solution for interoperability between private 5G sensor networks and OPC-UA-based industrial applications. Building on extensive research by S. Cavalieri on OPC UA interoperability [19,20,21], this study proposes a framework for mapping the YANG data models to OPC UA information models. The main aim of this research is to develop a framework to create OPC UA model design XML files from existing YANG models that UA model compilers can utilize to produce OPCA UA NodeSet2 XML files and other language files. Using a rigorous rationale, the authors analyzed the semantics of the YANG and OPC UA elements involved within the scope and established mappings between them. Based on the mapping, the authors proposed the OPC UA pyang plugin to convert the YANG file to an OPC UA model design XML file. pyang is an extensible validator and converter in python that enables developing plugins to convert YANG modules to other formats [22]. The major contribution of this work is that it offers thorough reasoning for the defined mapping through a semantic analysis of the involved elements and the development of a pyang plugin to support interoperability. The OPC UA pyang plugin has been released in the GitHub repository and is freely available [23]. The key finding of this research is that, from a practical standpoint, the proposal facilitates seamless interoperability between private 5G sensor networks and OPC-UA-based industrial applications without requiring modifications to the private 5G sensor network components. This interoperability, in turn, enhances the efficiency and reliability of industrial automation systems and their management, as existing industrial applications can be leveraged to manage and monitor private 5G sensor networks effectively.

The remainder of the article is structured as follows. Section 2 highlights the related work in the literature, the existing gap, and the need for this proposal. Section 3 provides an overview of YANG and OPC UA. Section 4 analyzes the semantics of YANG and OPC UA elements and defines the mappings between them. Section 5 describes the implementation of the OPC UA pyang plugin and demonstrates its validation using a case study example. Finally, Section 6 concludes the article. The abbreviations used in the article, the YANG and OPC UA XML code, and the mapping figures referenced in the article are included in the Appendix A. These elements are cross-referenced within the relevant sections of the article to ensure ease of reference and clarity.

## 2. Related Work

The existing literature on OPC UA interoperability with other protocols highlights numerous efforts to enable seamless communication in industrial environments. A substantial amount of research has explored the integration of OPC UA with Industry 4.0 reference architecture models, such as the asset administrative shell, relational databases, model-driven software development techniques, like UML, CIM, and QVT, as well as various domain-specific approaches, including smart grids, building automation, and semiconductor manufacturing [24]. Table 1 summarizes only the most relevant related work and the solution proposed in each of those studies.

The 5G Alliance for Connected Industries and Automation (5G-ACIA) outlines the necessary capabilities of private 5G network systems for industrial applications. It also details operational use cases that enable factory operators to carry out daily tasks without relying on 5G network operators [30]. In the 5G and Industrial Wireless Conference Stage [31], the expert discussed quality of service (QOS) mapping between OPC UA and 5G user plane traffic from a protocol perspective. The OPC Foundation and 5G-ACIA intend to integrate OPC UA with 5G systems [32]. Without interoperability support, enterprises are required to access various interfaces for network management and manually establish correlations between different interfaces. This research identified a significant gap in the need for interoperability and management of private 5G sensor networks using OPC UA systems. To realize these, interoperability between YANG and OPC UA information models is imperative as an interim step towards integration.

## 3. Background on YANG and OPC UA

This section aims to provide the reader with a basic overview of the fundamental technical concepts of YANG and OPC UA required to comprehend the article’s content.

### 3.1. YANG

YANG is a data modeling language commonly used in networks and device configurations to define data structures and relationships for network management protocols, such as NETCONF, which IETF created as a network configuration standard [33,34]. O-RAN Alliance Working Group 4 standardized the Management Plane (M-Plane) for managing the RU. It defines YANG data models for managing, configuring, and monitoring the RU. The specification defines two architectures for managing the RU. In the hierarchical model, the RU is managed entirely by DUs, whereas in the hybrid model, the RU can be managed by centralized network management systems as well as DUs [35]. O-RAN Alliance Working Group 10 and 3GPP standardized the operations and management of DU and CU in the O1 specifications [36]. O1 Interface specification for CU defines the interfaces between CU and the centralized network management system by defining the YANG models for management, configuration, and monitoring. O1 Interface specification for DU defines the interfaces between DU and the centralized network management system by defining the YANG models for management, configuration, and monitoring. The yang models for RU, CU, DU, and 5GC are maintained in the 3GPP repository [37]. The YANG data model defines the data tree that represents the hierarchical structure of defining components’ data. Figure 2 summarizes the YANG’s several built-in constructs, allowing data modeling and representation flexibility.

#### 3.1.1. Module

The module statement is identified by the module’s name, followed by a block of sub-statements that hold detailed module information [33]. The standard order of sub-element definitions within the module is header information, linkage statements, meta-information, revision history, and data definition statements.

Header information includes the YANG version, namespace, and prefix statements. The namespace statement within the module defines the unique qualifier for all of the statements defined within the module. The namespace and prefix statement are used in module definitions to differentiate the modules from each other. If the definitions from different vendors have the same names for YANG elements, naming collisions can be avoided.Linkage statements contain import and include statements. The import statement references the definitions from another YANG module within the current module using the prefix with a different namespace. The included statements are used for the definitions from another YANG module within the current module, and the current module namespace is used to reference those.Meta-information includes information related to the YANG definition, such as the organization name, description, and contact details.Revision history includes the revision statements, which track the history of changes to the YANG module and correlate to the different software versions of the managed device that support the YANG module.

#### 3.1.2. Data Definitions

Data definition statements include data type definitions, configuration and state definitions, and any Remote Procedure Calls and notifications supported by the device. They are the critical elements that define the device’s capabilities.

leaf is the simplest data definition, having at most one instance and no child data definition statements. leafs contain data values.leaf-list is otherwise the same as a leaf but can contain a list of unique leaves. A key refers to its value in the context of a leaf-list.container is a data dentition that is at most one instance and can hold no value but can have one or more child data definition statements, such as leaves or containers.list is otherwise the same as the container but can contain a list of unique containers. It is identified with one or more key leaves.grouping is data definition statements that can be used in multiple locations in the YANG model if grouped with a grouping statement. This enables the efficient reuse of definitions, thus reducing the modeler’s workload and decreasing the likelihood of error. When a part of a tree is used in multiple locations, making a change to it affects all instances of that structure.typedef statement: each leaf and leaf-list data definition statement includes a mandatory definition that specifies the format required for the data to be considered valid. YANG provides basic built-in types, such as string, enumeration, uint64, etc.; these data types can be extended by redefining the type definitions using typedef statements.The rpc statements define remote callable procedures with specified input arguments and output results. Remote Procedure Call (RPC) definitions can thus establish a comprehensive application programming interface (API) that can be efficiently used over the network.Notification statements can be used to specify a set of important events emitted by network functions. These notifications can contain complex information using the same rules as data definitions. Consequently, notifications can provide a valuable event-based interface for the state of network functions. Client applications can subscribe to these events to observe device state changes.

### 3.2. OPC UA

This section aims to provide the reader with the basic overview of fundamental technical concepts of OPC UA required to comprehend the article’s content. OPC UA is an industrial automation standard used to exchange information and configure and control industrial devices and applications from different manufacturers. Figure 3 defines the scope of OPC UA in industrial automation [16].

OPC UA primarily employs a client/server communication model based on standard TCP/IP, Web Sockets, and HTTP technologies. Figure 4 demonstrates the typical client/server communications in OPC UA [38]. The OPC UA framework provides a fundamental information model framework and a range of services for creating and sharing vendor-specific information in a standardized way. The OPC UA information models encapsulate the structure, behavior, and semantics of the OPC UA server. They consist of a network of OPC UA nodes that represent a variety of information types, including configurations, observability data, procedures/functions, and their interrelationships with other OPC UA nodes.

#### 3.2.1. OPC UA Address Space Model

The address space model serves as the OPC UA’s meta-model, laying the groundwork for OPC UA information models. Its main goal is to provide servers with a standardized way to represent objects to clients. Figure 5 summarizes all of the elements of the OPC UA address space model. The address space comprises a collection of nodes, each described by its attributes and references, which define the set of objects and their relational information [39]. Each OPC UA information model is identified using a namespace URI, and each OPC UA node is identified using an identifier. An OPC UA node belongs to a node class defined in terms of the attributes and references instantiated when a node is defined in the address space. Attributes are data elements that describe nodes, and OPC UA clients can access attributes using read, write, query, and subscribe functions. References establish connections of the OPC UA node to other OPC UA nodes. OPC UA defines node classes, which fall into three categories: node classes used to define instances, node classes used to describe types, and data types.

#### 3.2.2. OPC UA Device Model

The device model for the OPC UA server is established using one of the following methods: (a) leveraging the OPC UA standard-defined models, (b) leveraging the companion models, and (c) utilizing vendor-specific models. These device models are integrated into the OPC UA server and are accessible and manageable by any OPC UA client. OPC UA publishes a base information model, while many vendors publish companion information models, providing a unified view of devices regardless of underlying device variants.

#### 3.2.3. OPC UA NodeSet2 XML File

The OPC UA specifications define an XML schema (UANodeSet.xsd) for describing an OPC UA information model as an XML document, commonly referred to as NodeSet2.XML [40]. OPC UA SDKs can utilize NodeSet2.XML files to construct the OPC UA server address space either statically or dynamically. Any custom information model or companion specification must be provided in the official NodeSet2.xml format. This file contains all of the nodes and references between the nodes within the specific information model. The OPC Foundation has standardized the device models for different industrial devices and applications, and the same is maintained in the GitHub repository [41]. For private 5G network elements, manually developing NodeSet2.XML files is a cumbersome effort, and maintaining the same for every revision of the O1 specification will be tedious.

#### 3.2.4. OPC UA Model Design XML File

The structure of NodeSet2.xml is too intricate for any designer to create the files manually. As a result, the OPC Foundation has defined the UAModelDesign.XSD schema to represent the model design in a human-readable format using a simplified XML file structure [42]. A model design XML file must commence with a model definition to generate a model. There are various commercial and open-source UI-based designer tools available on the market [43,44]. These tools enable users to design the information model in the UI and then export the model design as an XML file. These model design XML files can then be converted into NodeSet2 XML files and any programming-language-specific files using the UA model compiler. The UA model compiler reads the Model.xml file, checks its consistency and integrity, and then generates the NodeSet2.xml files along with other required files [45].

## 4. Mapping YANG to OPC UA

The proposed mapping involves a two-stage approach. As depicted in Figure 6, the first stage involves the offline generation of the OPC UA information models from the 5G network systems’ YANG schema. The model generator takes the YANG schema as input and generates the mapped OPC UA information model files based on the mapping metadata. In the second stage, at runtime, the adapter converts the YANG model instances into corresponding OPC UA information model instances for the connected device. The primary motivation of this article is to propose a solution to address the first stage with the support of generating the OPC UA information models from YANG models. The second stage shall be addressed in future work.

The NodeSet2.xml structure is too complex for tools to generate, so the OPC Foundation has introduced the UAModelDesign.XSD schema to represent the model design in a more simplified XML file structure. This article presents a proposal for mapping YANG statements to OPC UA model design XML constructs, thus allowing the generated XML file to be converted into an OPC UA NodeSet2.xml file using the existing UA model compiler. The following sections offer a detailed proposal for mapping YANG constructs to OPC UA model design XML constructs.

### 4.1. YANG Module Mapping

The process of mapping YANG to OPC UA model design begins with the YANG module statement, which acts as the highest level of the YANG data model hierarchy. Each YANG module is then correlated to the model design in OPC UA, and the mapping process involves associating various elements of the module. Table 2 summarizes the mapping with reference to the Appendix A Section for a detailed visual representation of the mappings. The module name from YANG is mapped to the file name in the OPC UA model design. The namespace from the YANG model is mapped to the default XML namespace, the target namespace, and the target XML namespace in the model design XML file. The prefix statement from YANG is mapped to the namespace element prefix and the XML prefix attribute. For every import and included statements in YANG, the OPC UA model design XML file shall contain a namespace element with a prefix, an XML prefix, and a namespace URL. The organization statement in YANG specifies the name of the organization responsible for this module, while the contact statement provides contact information for the module, including the name, postal address, telephone number, and email address. Because these statements cannot be directly mapped to any OPC UA model design elements, a custom object type named module info is created. An object instance of this type is then mapped with these statement values. The revision statement is an essential part of the YANG schema providing detailed revision information and dates related to the module’s editorial history. However, it cannot be directly mapped to any OPC UA model design elements. To address this, a custom object type called revision info is created, and an object instance of this type is mapped to the revision statement values.

### 4.2. YANG Built-in Data Type Mappings

The YANG data definition statements leaf and leaf-list contain a mandatory data type definition statement that defines the type of data the elements contain. Table 3 describes the built-in data types supported by the YANG model and the proposed mapping of OPC UA built-in data types. In cases where no equivalent data type is defined in the OPC UA model, either the custom object type is created or an equivalent data type that does not alter the data value definition is chosen.

### 4.3. YANG Data Definition Mapping

Data definition statements in the YANG Model represent the entity’s configuration or state data, as well as the methods and notifications it supports. Table 4 summarizes the data definition mapping approach with reference to the code and images defined in the Appendix A Section. The module’s data definition statements leaf, leaf-list, container, grouping, and list in YANG are mapped to the object type, variable type definitions, and other elements in the OPC UA model design XML file.

leaf statement: leaf is the simplest data definition in YANG, which is, at most, one instance with data values and has no child data definition statements. The leaf statement defines a scalar variable of a particular built-in or derived type. The config statement in the data definition identifies whether the definition represents the element’s configuration or the state data. Any data definition defined as a config to true can be edited. Otherwise, it can only be observed. If the leaf statement is defined with config as “true”, then in OPC UA, it is mapped to a variable node, and if it is described as “false”, then it is mapped to a property node.leaf-list statement: leaf-list statement is used to define an array of particular data types. leaf-list contains a list of unique leaves with a key that refers to their value. Like the leaf statement, the leaf-list is mapped to a property node or a variable node in OPC UA with the ValueRank attribute of the node set to OneDimension and the ModellingRule attribute set to “ExposesItsArray”.Container statement: container is a data definition that wraps one or more child data definition statements. The OPC UA address space model does not provide a node for representing the container; hence, the container is emulated by generating a custom ObjectType and defining an object instance to the custom object type.Grouping statement mapping: the grouping statement defines a set of nodes that can be assembled into reusable collections and instantiated with the uses statement. In OPC UA, grouping statements are mapped to a custom object type and instantiated using the object declaration of the type.List statement: list contains one or more child data definition statements, including a grouping of leaf or container entries, with each uniquely identified with one or more key leaves. The list statements are mapped to an ObjectsFolder reference by custom object types. Also, the holder object is created in the objects folder so that the list entries can be added programmatically at runtime.Typedef statement mapping: typedef statement allows for the creation of a new derived data type by redefining the base type. The base type can be the YANG built-in type or a custom type. Typedef statements are mapped to DataType definitions in OPC UA.RPC statement mapping: the rpc statement defines a YANG RPC operation. It contains a block of sub-statements that holds detailed input and output nodes. The operation name, input, and output parameters are modeled using YANG data definition statements. The RPC statement is mapped to the method node in the OPC UA model design. The notification statement defines a NETCONF notification. YANG data definition statements model the notification content. The notification is mapped to Custom Event Type objects in the OPC UA model design.

## 5. Implementation and Validation

A YANG to OPC UA model design generator is developed as a plugin on top of existing open-source software called pyang [22]. The logic of the OPC UA pyang plugin for the generation of the OPC UA model design file is based on the YANG to OPC UA mapping explained in previous sections. The generated XML output conforms to the schema defined in the UA Model Design.XSD and then passed to the existing OPC foundation model compiler to generate the OPC UA NodeSet2 XML files and different-language SDK files, as depicted in Figure 7. The authors published the opc-ua pyang plugin and its software implementation, the sample YANG and OPC UA files, and the documentation on how to use the plugin to generate OPC UA NodeSet2 files in their GitHub repository [23].

pyang is a YANG validator, transformer, and code generator command line tool written in Python, XSLT, and shell scripts. It can be used to validate YANG modules for correctness and to transform YANG modules into other formats using custom plugins. The plugin can be invoked using the “-f” option in the command line by specifying the plugin name. The OPC UA plugin developed as part of this article is named “opc-ua”. The command line syntax used to execute the plugin is as follows:



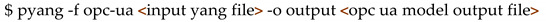



The plugin name “opc-ua” must be specified using the option “-f” followed by the input YANG file, and then the “-o” option must be used to specify the OPC UA model design XML file name. The OPC foundation model compiler can generate the OPC UA NodeSet2 XML files and other language source files from the generated OPC UA model design XML file, and the command syntax for the same is shown below:



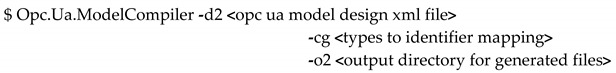



Option “-d2” specifies the path to the model design file generated using the pyang opc-ua plugin. Option “-cg” is used to specify the path to the CSV file, which contains the unique identifiers for the types defined in the UA information model. Option “-o2” specifies the output directory for the generated files.

### 5.1. Case Study

To provide a clear understanding of the proposed solution, this subsection presents a single and very simple case study as an example. The subsequent validation subsection provides a comprehensive overview of the parameters considered for validation, along with the detailed results. The YANG file used in this case study is available in the author’s GitHub repository [46]. The YANG module “simple-example” contains a container definition called “bag”, a grouping definition called “my-grouping”, and another container named “example-container” uses this grouping definition. Also, the module contains other definition statements, such as typedef, leaf-list, and rpc definitions. The YANG “simple-example.yang” is passed as an input file to the pyang opc-ua plugin using the following command:



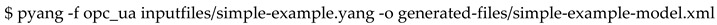



The generated OPC UA model design file is maintained in the author’s GitHub repository for reference [47]. The generated file contains the ModelDesign root XML element with namespace prefix and namespace declarations as defined in the YANG. It encompasses child node namespaces, ObjectType nodes for each grouping statement, variable nodes for leaf and leaf-list statements, and ObjectsFolder type for list statements. The generated simple-example-model.xml is then passed to the OPC UA model compiler to generate the OPC UA NodeSet2 XML files using the following command:



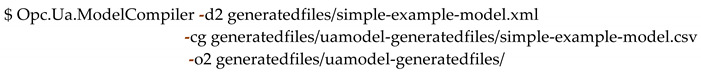



The model compiler generated multiple files, such as NodeSet2.xml, Types.xsd, PredefinedNodes.xml, and C# template files. These are maintained in the author’s repository for reference [48]. To validate the generated NodeSet2 XML files, the model is loaded into the FreeOpcUA python-opcua server [49]. Then opcua-client-gui is connected to the server to browse through the information models and RPCs [50]. Figure 8 shows the screen capture of the OPC UA GUI client with the simple example NodeSet2 XML files loaded in the OPC UA server. In the GUI client, objects defined in YANG, such as Bag, ExampleContainer, and ListOfPorts, are visible.

### 5.2. Validation of Generated Model Design File

To demonstrate interoperability, several YANG files related to 5G RU, CU, DU, and 5GC were considered [37]. The generated OPC UA model design files were validated to ensure the mappings proposed cover different positive and negative scenarios. Table 5 summarizes the performance of the OPC UA plugin with respect to different categories of YANG files used during the generation of the OPC UA model file. The plugin and the generated files are validated for correctness, interoperability, efficiency in terms of generation time, handling large data models, supporting multiple files at the same time, adaptability to new standards, error reporting, and resilience.

Correctness of the model mapping: the generated files are analyzed semantically to ensure that the YANG data models are accurately mapped to corresponding OPC UA information models without semantic loss. Structural integrity is validated to ensure that the hierarchy and relationships in YANG models (like containers, lists, and leafs) are preserved accurately in the OPC UA structure (like objects, variables, and methods).An interoperability test is conducted to test if the generated OPC UA models can be seamlessly loaded into OPC UA server instances and if models can be accessed either using OPC UA client or OPC UA GUI instances.Generation time: measures how efficiently the plugin processes YANG models and generates the corresponding OPC UA models.Handling large data models: tests the plugin’s ability to manage increasingly large or complex YANG data models. Verifies that the plugin performs well with deep and broad data hierarchies and numerous variables or configurations.Support for multiple YANG modules: validates whether the plugin can handle multiple YANG modules and revisions while maintaining flexibility to incorporate various configurations or extensions.Adaptability to new standards: checks whether the plugin can adapt to new or updated versions of YANG and OPC UA standards as they evolve.Error reporting: ensures that the plugin provides clear and detailed error messages when issues arise during the model generation process, such as invalid YANG models or unsupported features.Resilience: tests the plugin’s robustness against incomplete or erroneous input data. It should either gracefully handle errors or provide meaningful feedback for corrections.

The generation time is predominately taken for parsing the YANG file and generating the OPC UA model design XML files, as these involve disk IO operations. The actual mapping logic in the plugin takes on an average of around 30% of the total time. Also, the variability in time across different models is primarily driven by the number of statements. During testing, it was observed that if the YANG model contains import or includes statements referencing other YANG models, which is the case for the majority of 5G YANG models, the parent YANG models must first be converted into OPC UA model design XML files using the OPC UA plugin. Then, all of the generated OPC UA model design XML files must be passed as input files to the UA model compiler so that the compiler can generate a single NodeSet2 XML file linking all of the files. In rare scenarios, the UA model compiler was observed to fail with internal errors; however, most of the time, it successfully generated the required files. Therefore, the results related to supporting multiple YANG modules are marked as partial in the results summary table. Another issue identified with the import and included statements was that the OPC UA plugin generated duplicate namespace URIs in the OPC UA model design XML files. This issue was worked around by adding a YANG prefix to the URIs. The OPC UA plugin supports parsing only one input YANG file at a time, which led to the observed issue of duplicate namespace URIs. Modifying the plugin to read and parse the imported/included YANG files recursively will ensure that namespace URI collisions can be avoided; this enhancement is planned for future updates to the plugin. Figure 9 shows the OPC UA information model generated from the 5G RU YANG model, which was loaded into the OPC UA server and viewed through the OPC UA client GUI. Because the runtime instance of the YANG to the OPC UA adapter is yet to be developed, the FreeOpcUA client application was used to modify the temperature attributes of the QoS object in the RU information model. The exact change was successfully reflected in the GUI application when it subscribed to the data change events.

## 6. Conclusions

The digital factory is a crucial enabler for Industry 4.0 to continuously innovate to create new revenue streams, reduce costs, increase efficiency, and build eco-friendly industries. The integration of a private 5G network with sensor networks represents a significant advancement in wireless communication, offering new opportunities for various applications. The solution proposed in this research achieves interoperability between YANG and OPC UA, the two significant technologies utilized in network management and inter-device communications. A substantial contribution of this work is that it offers thorough reasoning for the defined mapping through a semantic analysis of the involved elements and the development of a pyang plugin to support interoperability. The OPC UA pyang plugin has been released in the GitHub repository and is freely available. The proposed mapping method in the research allows YANG data model-based network devices to operate independently without the need for modification, making it easy to integrate them with any OPC UA industrial applications for network management. This research focuses on transforming the YANG data model to an OPC UA design model file. It is also feasible to go in the opposite direction from the OPC UA to the YANG data model. However, this would require the development of an extensive metadata model covering all OPC UA nodes. The authors are currently developing a YANG/NETCONF to OPC UA adapter to ensure interoperability at runtime. This adapter will support the NETCONF protocol for communication with devices on the southbound side and the OPC UA protocol for communication with OPC UA applications on the northbound side. This enables seamless interoperability without the need for modifications to either the network devices or OPC UA client applications. This research is original, as no other contributions are present in the current literature on the same subject, and it paves the way for the widespread adoption of 5G sensor networks in industrial automation.

## Figures and Tables

**Figure 1 sensors-24-06231-f001:**
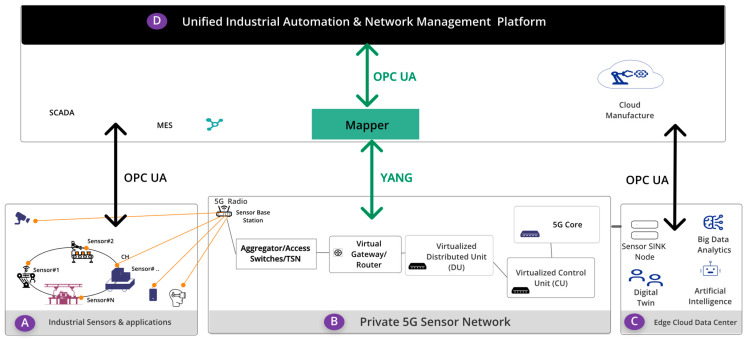
Unified management interface for 5G sensor network.

**Figure 2 sensors-24-06231-f002:**
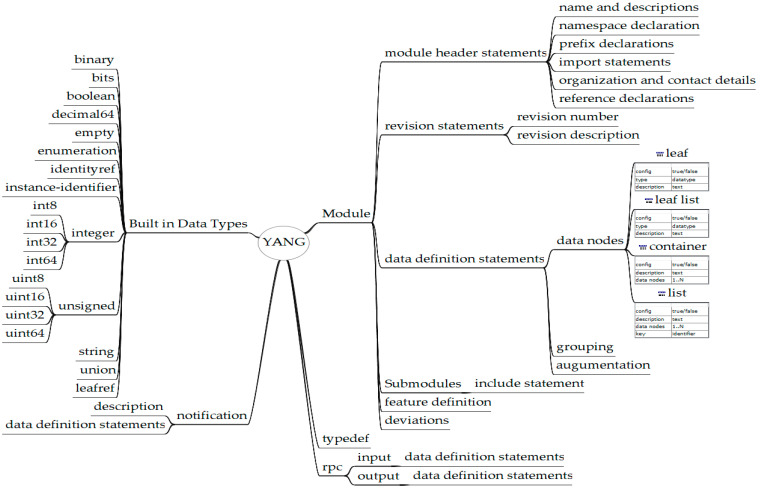
YANG language construct.

**Figure 3 sensors-24-06231-f003:**
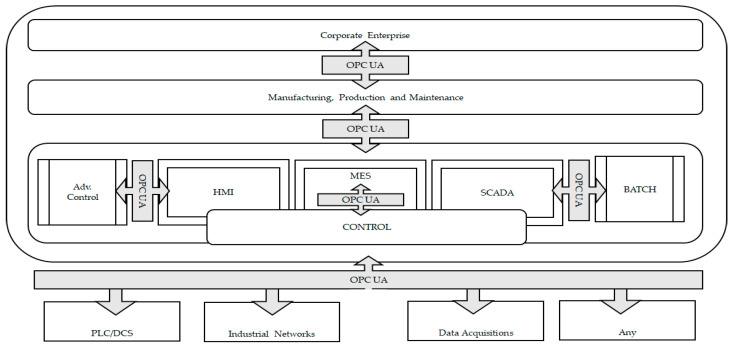
OPC UA’s scope.

**Figure 4 sensors-24-06231-f004:**
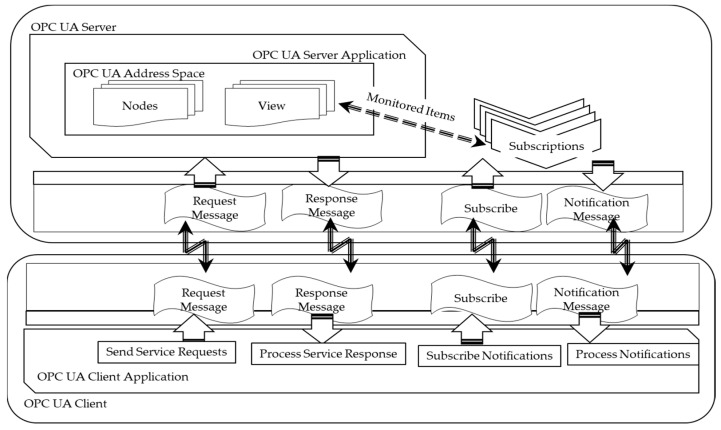
OPC UA client–server communication.

**Figure 5 sensors-24-06231-f005:**
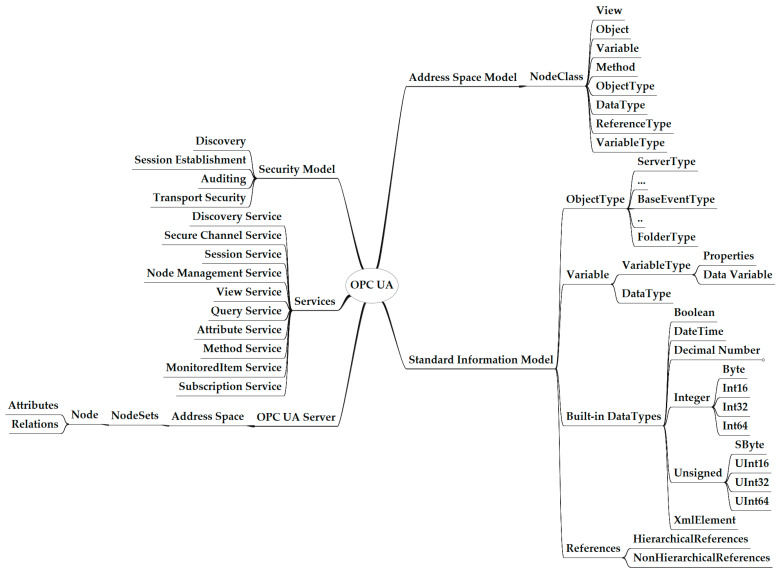
OPC UA model.

**Figure 6 sensors-24-06231-f006:**
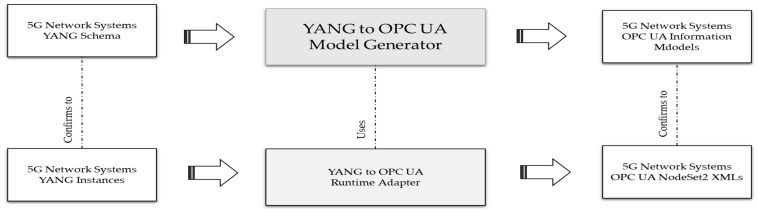
YANG to OPC UA model mapping.

**Figure 7 sensors-24-06231-f007:**
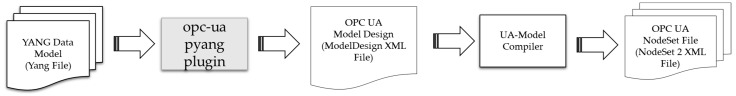
YANG schema to OPC UA model generation.

**Figure 8 sensors-24-06231-f008:**
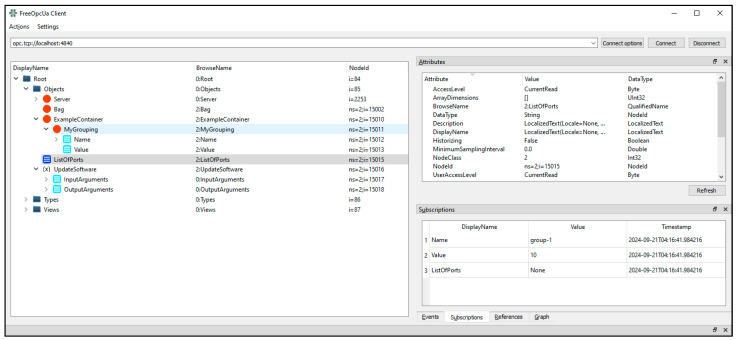
Generated simple example of an address space model loaded in the OPC UA server.

**Figure 9 sensors-24-06231-f009:**
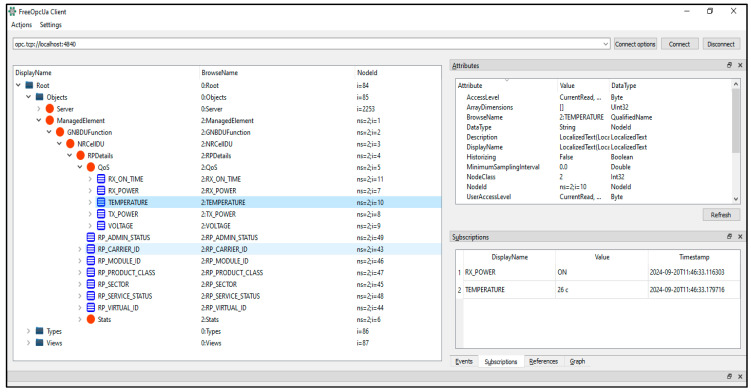
Generated 5G RU OPC UA information model loaded in the OPC UA server.

**Table 1 sensors-24-06231-t001:** Existing interoperability solutions between OPC UA and other models/protocols.

Data Model/Protocol	Interoperability Solution Proposed	Reference
Digital Twin Definitions Language (DTDL)	Introduces the solution of mapping DTDL to OPC UA information, thus allowing each DTDL element to be represented by a corresponding OPC UA element.	[19]
One Machine to Machine (OneM2M)	Proposes interworking between OPC UA and OneM2M, thus enabling access to information managed by OneM2M-based systems/platforms in OPC-UA-based applications.	[20]
Open Connectivity Foundation (OCF)	Discusses automatically generating the OPC UA information models from high-level OCF design models.	[21]
Unified Modeling Language (UML)	Describes an approach for transforming OPC UA to UML, where the authors analyzed the semantics of OPC UA elements and mapped them to corresponding UML elements.	[25]
Automation Markup Language (AutomationML)	Examines the creation of OPC UA information models based on existing AutomationML data, highlighting the analogies between AutomationML and the OPC UA information model.	[26]
System Modeling Language (SysML)	Proposes automatically generating OPC UA information models from high-level SysML design models.	[27]
Robot Operating System (ROS)	Suggests a local cloud-based approach to achieve interoperability between ROS and OPC UA by integrating them with the eclipse arrowhead framework.	[28]
Common Information Model (CIM)	The smart grid platform was developed to comply with CIM and OPC UA standards and ensures secure interoperability among numerous legacy systems.	[29]

**Table 2 sensors-24-06231-t002:** YANG module statement mappings to OPC UA model design XML constructs.

YANG Statement	Mapped OPC UA Model Design XML Constructs	Visual Representation
module <module-name>	ModelDesignFileName.xml	Figure A1
namespace <namespace> prefix <prefix> import <module-name> { prefix <prefix>; }	<ModelDesign … TargetNamespace = “namespace” … TargetXmlNamespace = “namespace-name”> <Namespace Name = “name” Prefix = “prefix” … XmlNamespace = “xsduri” XmlPrefix = “xmlprefix”>uri</Namespace>	Figure A2 and Figure A3
organization contact description reference	Custom object type definition named “moduleinfo”	Figure A4 and Figure A5
revision <name> { reference; description }	Custom object type definition called “revisioninfo”	Figure A6 and Figure A7

**Table 3 sensors-24-06231-t003:** YANG data type to OPC UA built-in data type mapping.

Yang Built-in Data Type	Description	Mapped OPC UA Built-in Data Type
binary	Any binary data	Not supported directly; OPC UA ByteString type used to map binary data
bits	A set of bits or flags	BitFieldMaskDataType
boolean	True or “false”.	Boolean
decimal64	64-bit signed decimal number	Double
empty	A leaf that does not have any value	BaseObjectType
enumeration	Enumerated strings	EnumValueType
int8	8-bit signed integer	SByte
int16	16-bit signed integer	Int16
int32	32-bit signed integer	Int32
int64	64-bit signed integer	Int64
string	Human-readable string	String
uint8	8-bit unsigned integer	Byte
uint16	16-bit unsigned integer	UInt16
uint32	32-bit unsigned integer	UInt32
uint64	64-bit unsigned integer	UInt64
date-and-time	Data and time	DateTime
Union	Represents a value that corresponds to one of the member types	union

**Table 4 sensors-24-06231-t004:** YANG data definition statements to OPC UA model design XML constructs mapping.

YANG Statement	Mapped OPC UA Model Design XML Constructs	Visual Representation of the Mapping
leaf { … config false; …}	<Property SymoblicName = “…” …>	Figure A8
leaf { … config true; …}	<Variable SymoblicName = “…” …>	Figure A9
leaf-list … {…}	<Variable … ModuleRule = “ExposesItsArray” …>	Figure A10
container …{…}	<ObjectType SymoblicName = “…”…>	Figure A11
grouping …{…}	<ObjectType SymoblicName = “…”…>	Figure A12
list …{…}	<ObjectType SymoblicName = “…”…>	Figure A13
typedef <customtype> {..}	<DataType SymoblicName = “…”…>	Figure A14
rpc …{…}	<Method SymoblicName = “…”…>	Figure A15

**Table 5 sensors-24-06231-t005:** Validation results of opc-ua plugin against different categories of YANG files.

YANG File Category	Correctness	Interoperability	Generation Time	Handling Large Models	Multiple Yang Modules	Adaptability to New Standards	Error Reporting	Resilience
Switch YANG	Yes	Yes	5 10 s	Yes	Partial	Yes	Yes	Yes
CU-CP	Yes	Yes	10–25 s	Yes	Partial	Requires code change	Yes	Yes
CU-CU	Yes	Yes	10–15 s	Yes	Partial	Requires code change	Yes	Yes
DU	Yes	Yes	10–20 s	Partial	Partial	Requires code change	Yes	Yes
5G Core	Yes	Yes	20–50 s	Partial	Partial	Requires code change	Partial	Partial
RU	Yes	Yes	10–20 s	Yes	Partial	Requires code change	Yes	Yes

## Data Availability

Data are contained within the article.

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
