# Peer review of "Towards a Unified Management Interface for 5G Sensor Networks: Interoperability between Yet Another Next Generation and Open Platform Communication Unified Architecture"

_sensors, 2024, doi:10.3390/s24196231_

Round 1

Reviewer 1 Report

Comments and Suggestions for Authors

This manuscript deals with the interoperability between two standard protocols of data exchanges, and system communication more broadly, namely OPC UA and YANG, and in particular from the latter to the former. To that purpose, the authors analyze the XML file semantics of both protocols, and propose a Python plugin, namely OPC UA pyang, to "translate" a YANG file into an OPC UA one (cf. ref [16]).

This research is interesting, and is worth being published. From a technical point of view, the article is fine with me in its current form, but I think that it is worth being improved from a "format" point of view. Indeed, I have got the feeling that the authors want to say too many things, and share their (great) work, but this causes the article to be confused -- and confusing -- here and there, and a little disjointed, in particular due to the use of many large images, and many very short subsections.

Although, as a reviewer, I will not go against the publication of the article, for it is interesting and is worth being published, I strongly recommend the authors work on that point, and improve the fluidity of their article. To that end, the authors will find my pieces of advice below, that I recommend the authors take into account in order to improve the quality of their article.

Abstract:
A little long, some elements may be removed without diminishing the relevance of the abstract. Also, I think it would be great if the authors may include the main findings of their study.

1. Introduction
The introduction is interesting, but much too long, with several elements which are repeated several times, all along the core text: I suggest that the authors summarize the repetitive, non-essential information. Additionally, Figure 1 is not properly used: it is interesting, but not explained, and is barely linked to the main matter by a simple "As shown in Figure 1". Finally, it may be a personal preference, but I would have provided the article content in a separate paragraph, i.e. from "The remainder of the article" (l.137) to "Section 8 concludes this article" (l.143).

2. Related Work
This section is short, and I do not always see a direct link between the references and the proposal, but it is not surprising, since, according to the authors, the present manuscript proposes a fully innovative approach, never published (nor investigated?) in existing literature. As such, it may be hard to build a true state-of-the-art on that specific topic, thus that's fine with me.
However, in the table, please, use only one tense, either simple present or simple past (e.g. "It introduced" for [11] and "Discusses" for [13]).

3. YANG & 4. OPC UA
These sections are interesting, but are unclear. They appear to be long, although they are actually not that much; in my opinion, the problem comes from the images (source codes), which take up too much space here.

I suggest fusing these two sections, and separating the resulting section into two subsections, the first for YANG, and the other for OPC UA. In each, I would avoid having up to 7 subsubsections, some of which are very short (e.g. subsection 4.6, which is only 3 line-long): as such, I would merge the core text when possible (for instance, sections 3.2 and 3.3 could easily be merged), and list the main elements in tables. For instance, the different statements explained in section 3.1 could be summarized into a simple list or table, which would be as efficient as the present text, but also more readable; for example, section 3.2 is much better from this point of view.

Additionally, I would avoid including as much source code into the core text, I would move them to the very end of the article, as appendices, for these lines are not mandatory to understand the role of each element discussed, they only serve as very general example of use. It would be good to keep them, but I have the feeling that they make the text heavier, and harder to read, for the paragraphs are severely split by these large images.

Also, I suggest not including so much detailed descriptions of elementary elements of YANG and/or OPC UA (in particular, those which are common to almost any OOP language, such as what is a data type, a variable or even an object method). For instance, in section 4.2, the authors provide many details on the different objects, their types, roles, etc.; this is also noticeable in section 4.3, in which the authors define what is an "OPC UA Device model", but, beyond the intrinsic interest one may have, I do not see the benefit in providing as many details when they do not really serve the rest of the study. Albeit interesting, the authors may consider that the reader is familiar with these definitions, especially someone who would look for a framework to translate a YANG file into an OPC UA. As a consequence, except if this is mandatory for the rest of the study (I mean, the way that the authors propose to translate these elements), I suggest simplifying these sections, and keeping only the bare essentials to understand the study, for this research article should not be a technical manual explaining how two standard protocols work. If the authors really want to keep these pieces of information, I would suggest moving them to more detailed appendices.

Finally, please, do not provide screenshots of source codes, this is barely readable; instead, prefer including well-formatted (and colored) source codes. For instance, the authors may have a look at this Stack Overflow discussion (that I am not involved in): https://stackoverflow.com/questions/3252098/what-is-the-best-way-to-insert-source-code-examples-into-a-microsoft-word-docume.

5. Mapping YANG to OPC UA
This section is well-written and explained, but the images are too invasive: they are large and numerous, what diminishes the fluidity of the text. For instance, on page 14, there are two very short subsections, 5.2 and 5.3, each comprising 2-4 lines of text, plus a large image each: it is a little messy imo. These images are practical examples, illustrating the surrounding text, and that's fine; however, in the actual form, they do not really help the reader, for they are too large, not always consistent (e.g. the font size is clearly not the same between Figures 14 and 15), what greatly hampers their benefit. I suggest replacing the images by much simpler (and smaller) tables, each row indicating the equivalence between the YANG and OPC UA statements (e.g. "module <module-name>" and "<ModelDesignFileName>" in Figure 13), like Table 2 for instance, in which the YANG and OPC UA built-in datatypes are linked with much more clarity imo.

Nevertheless, I want to emphasize that these images are good, and illustrative (maybe too heavy nonetheless, perhaps the non-absolutely necessary statements could be removed, and replaced by something like "[...]" in the figures, in particular for OPC UA), thus I do not think that they should be removed. I think that moving these images at the very end of the article, as a separate appendix providing a true example of application, could be beneficial, and more readable. Also, I recommend adding small borders to the images containing source code, so as to make them easily distinguishable from the core text.

Also, I think that several subsections could be merged without diminishing the clarity of the proposal, e.g. sections 5.2 and 5.3.

Finally, in subsection 5.8, since the subsubsections (5.8.1, 5.8.2, etc.) are short (except the large images), I suggest replacing these subsubsections by a simple list, using simple bullets to separate the different items (leaf, leaf-list, container, etc.). Additionally, in the illustrating images, much of the text is not necessary, if I understood well, thus it could be removed without impacting their meaningfulness. For instance, in Figure 20, a simple table of two rows would be enough, since, if I am not wrong, the purpose of the image is to link "config false" with "Property" on the one hand, and "config true" and "Variable" on the other hand; as a consequence, all the (numerous) additional statements, albeit illustrative, are superfluous, and greatly diminish the fluidity of the text. Notice that I understand, and appreciate, the purpose of these images, as concrete examples of equivalence, but I think that they could simply be moved to an appendix for instance, to make the text lighter to read.

6. YANG to OPC UA Model Mapper
This section is too short to be a true section (I mean, 5 lines + 1 image). I recommend moving the section at the very beginning of section 5, as an overview of the proposed methodology. Else, nothing to report.

7. Implementation and Validation
In the section introduction, the authors are repeating themselves in the text before and after Figure 27: I recommend simplifying the core text, and keeping only the necessary (in practice, I think that the paragraph below Figure 27 may be safely removed).

Also, Figure 28 is unclear. I suppose that they are command lines to operate the "translation" from YANG to OPC UA using the pyang module, but this is not clearly stated. Please, introduce these command lines, and the figure more broadly. By the way, I think it would also be much better to include these command lines in the core text, not as an image.

Regarding Figures 29 & 30, I recommend explaining what the authors want to represent with these YANG & OPC UA file examples: it would be much more representative to provide the reader with a human-readable example, in my opinion. Indeed, in their current form, these two images are just barely readable examples of XML source code.

Finally, in subsection 7.2, I recommend the authors expand their validation. Indeed, in the current form, the validation stage of their study is only a 7-line long paragraph, which is not representative of their work. Please, provide examples, where the pyang module proved to be correct or wrong, what are the implications of these results (e.g. time saving? human workforce saving? other?), etc.

Notice that I personally have no problem with dealing with a very simple, academic example, I just recommend the authors go deeper in their analysis of their results.

8. Conclusion
Nothing to report in particular, expect that I would have not written "Industries" (line 683) with an uppercase "I" (maybe authors wanted to write "Industry 4.0", and then changed thereafter, I don't know). Also, I recommend including the link to their GitHub repository, either in the core text, as a footnote, or just as a reference

Reviewer 2 Report

Comments and Suggestions for Authors

1. The title of the manuscript, "Towards a Unified Management Interface for 5G Sensor Network: Interoperability Between YANG and OPC UA," suggests a strong focus on the 5G interface. However, the core of the paper (Chapters 5, 6 and 7) focuses on the conversion between the OPC UA model and the YANG model, with very little reference to anything related to the 5G interface. In fact, the word ‘5G’ does not appear once in these chapters. This creates a mismatch between the titles and the actual research focus. I suggest either revising the title to better reflect the main contribution of the paper, which is centred around interoperability between the OPC UA and YANG models, or expanding the discussion in these chapters to include relevant 5G interface topics to be consistent with the current title.

2.The introduction provides an overview of private 5G sensor networks and their significance in industrial automation. However, it lacks appropriate and up-to-date references to support several key points. For example, when discussing the benefits of private 5G sensor networks, more specific and relevant information from recent literature should be used to support the claims. Citing authoritative studies or industry reports would enhance the credibility of the discussion and give readers a clearer picture of the current state of research.

 3. Some figures (e.g., Figures 19, 21, 22 ...) and tables are not clearly referenced in the text, and their explanations are insufficient. The authors should ensure that all figures and tables are well-integrated into the discussion, with detailed captions that explain their relevance to the research.

4.There are several formatting and typographical errors in the manuscript that affect its readability and professionalism. For example, there are instances of double spaces in lines 44 and 89, and a number of other minor typographical errors throughout the text. It is important to proofread the document carefully to correct these problems and ensure consistency in spacing, punctuation, and formatting.

5. In Section 7.1, the manuscript states, "To demonstrate interoperability, several case studies involving different 3GPP YANG files were considered, and the generated OPC UA model design files were validated to ensure the mappings proposed cover different scenarios." However, the manuscript does not provide any details or examples of these case studies or validations. The lack of specific examples and results weakens the impact of the claimed findings. I recommend the inclusion of specific case studies, detailed descriptions of the validation process, and results that support the claims made. Without these, the manuscript lacks the necessary evidence to support the proposed solutions.

6. In Figure 30, the manuscript presents the OPC UA model design using an XML file. To enhance readability and clarity, it would be beneficial to display the XML file with color-coded syntax highlighting, making it easier for readers to distinguish different elements and attributes. In addition, a practical understanding of the proposed solution would be greatly enhanced if the designed OPC UA model could be instanced on an OPC UA server. The model can then be accessed and explored using tools such as UaExpert or similar platforms, giving the reader a clearer visualisation of the converted OPC UA model. Providing such an implementation will enhance the paper by giving the reader a more interactive and verifiable view of the results.

Round 2

Reviewer 2 Report

Comments and Suggestions for Authors

Thank you for addressing my comments, I am satisfied with the revisions and recommend acceptance.